# Effect of Selected Factors on the Bending Deflection at the Limit of Proportionality and at the Modulus of Rupture in Laminated Veneer Lumber

**Adam Sikora, Tomáš Svoboda \*, Vladimír Záborský and Zuzana Gaffová**

Department of Wood Processing and Biomaterials, Faculty of Forestry and Wood Sciences,
Czech University of Life Sciences in Prague, Kamýcká 1176, Prague–Suchdol 16521, Czech Republic;
sikoraadam@seznam.cz (A.S.); zaborsky@fld.czu.cz (V.Z.); gaffova@fld.czu.cz (Z.G.)

**\*** Correspondence: tomassvo@seznam.cz; Tel.: +420-608-321-941

**Abstract:** The deflection of a test material occurs under bending stress that is caused by force. In terms of plasticity and elasticity, the deflection can be quantified at two main areas, which are the limit of proportionality and the modulus of rupture. Both of these deflections are of great importance in terms of the scientific and practical use. These characteristics are particularly important when designing structural elements that are exposed to bending stress in terms of the size of the deflection in their practical application. This study analyzed the effect on the size of the deflection at the limit of proportionality and at the modulus of rupture. Wood species (*Fagus sylvatica* L. and *Populus tremula* L.), material thickness (6 mm, 10 mm, and 18 mm), non-wood component (glass and carbon fiber), position of the non-wood component in the layered material (up and down side with respect to the loading direction), and adhesive used to join the individual layers (polyurethane and polyvinyl acetate) were the observed factors. Glass fiber reinforcement proved to be a better option; however, the effect of correctly selected glue for individual wood species was also apparent. For the aspen laminated materials, polyurethane adhesive (PUR) adhesive was shown to be a more effective adhesive and PVAc adhesive was better for the beech-laminated materials. These results are of great importance for the production of new wood-based materials and materials were based on non-wood components, with specific properties for their intended use.

**Keywords:** technological and product innovations; cyclic loading; laminated wood; deflection at the limit of proportionality; deflection at the modulus of rupture; wood-processing industry performance

## 1. Introduction

The composition design and production of Laminated Veneer Lumber materials is primarily focused on the intended use of the material. Therefore, it is necessary to know the proposed composition under the influence of various factors in order to create such a material.

The basic component of LVL materials is wood component, however non-wood component can be also used. The single layers of LVL material could be modified or unmodified. The modification can be carried out in a variety of ways, such as high temperature, chemical, impregnation, pressure, microwave irradiation, lamination, etc. The possibilities of increasing the strength and the stiffness of wood lie in its combination with non-wood components at various material bases [1]. These are steel, glass, and carbon fibers in building materials [2–4]. Authors [5] conducted research with the effect of embedded carbon fiber fabric in the bonded joint of five-layer plywood to increase the flexural strength.

Flexibility is a technical property of wood that expresses its ability to bend. It is a material property that may be classified as either a positive or negative characteristic, depending on the purpose. In the

case of materials that are intended for bending, this property is desirable [6–8], whereas it is undesirable in materials that are intended for construction [9,10].

Bending is defined as a torque that acts on a material perpendicular to the cross section, which results in normal and tangential stress that causes stress through bending or twisting [11]. Each change in a beam subjected to bending is the result of work that is directly dependent on the force that is used and the resulting deformation [12,13]. The range of external forces doing the work changes in bending within the maximum boundary. The internal forces that are caused by deformation are also displaced [14]. Potential energy accumulates in an elastically deformed object, and then the energy is converted into work that is consumed after release to enable the object to return to its original shape [15–17].

The deflection at the limit of proportionality of wood is characterized as the boundary after which deformation becomes elastic over time and plastic. It is necessary to use the force at the limit of proportionality to achieve deflection at the limit of proportionality [18]. Up to this point, the wood is loaded with a force that only causes elastic deformation and it is therefore only flexibly loaded [19–21]. This deflection can be achieved without any apparent permanent deformation of its size or shape. The deflection at the modulus of rupture is characterized as the deflection when the material breaks. It is necessary to use force at the modulus of rupture to achieve this deflection (Figure 1) [20,22,23].

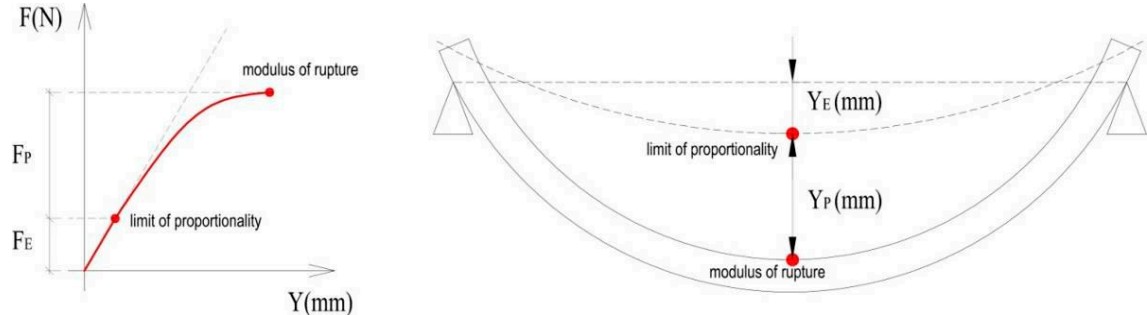

**Figure 1.** Force-deflection diagram of the bending stress and determining the limit of proportionality and the modulus of rupture [24].

The bending characteristics of the wood can be improved by reinforcing it with non-wood components on a different basis [1,25]. Carbon fibers, aramid fibers, basalt fibers, fiberglass, and polyvinyl alcohol (PVA), and others, are suitable non-wood components that can be used for wood reinforcement [26–28]. In addition to synthetic fiber reinforcement, natural fiber reinforcement may also be used, e.g., in the form of a nonwoven fabric. Experimental research has shown that correctly locating the reinforcement component has the potential to increase the bending characteristics of laminated wood in the case of application on the stressed tensile zone [21,29,30].

The literature points to the influence of surface structure on the properties of the bonded surface joint [31,32]. In view of this, the selection of a suitable adhesive for joining wood-based materials and non-wood components is very critical [33]. Wood itself is a hygroscopic material and, as such, it continually absorbs and releases moisture, which, among other things, causes a change in size. These changes can cause the adhesive to separate from the wood component. This effect can be avoided by appropriately selecting an adhesive for a specific bonded element [34].

The objective of this study was to determine the effect of the composition of laminated wood (wood species, type of non-wood component, position of the component in the structure, thickness of the material, and the type of adhesive used for bonding individual layers) on the deflection characteristics of the force-deflection diagram (deflection at the limit of proportionality and at the modulus of rupture).

## 2. Materials and Methods

### 2.1. Material

For the experiment, layered beech (*Fagus sylvatica* L.) and aspen (*Populus tremula* L.) wood (Polana, Slovakia), with a non-wood reinforcing component (glass and carbon fibers), were used. The test specimens were made with three thicknesses of 6 mm, 10 mm, and 18 mm, with a lamella width of 35 mm and length of 600 mm. The test specimens were divided into 48 test groups, with respect to the thickness and species of wood component, the type of non-wood component, and the position of the non-wood component relative to the load direction. Figure 2 shows the categorization of the test specimens. The individual wood and non-wood components were glued while using two types of adhesive, which were single-component waterproof polyvinyl acetate adhesive (PVAc) (AG-COLL 8761/L D3, EOC, Oudenaarde, Belgium) and single-component polyurethane adhesive (PUR) (NEOPUR 2238R, NEOFLEX, Madrid, Spain). Table 1 lists the detailed parameters of these adhesives. The test specimens were climatized at a 12% equilibrium moisture content (EMC) in a climatic chamber (Binder ED, APT Line II, Tuttlingen, Germany) at set relative humidity (65%) and temperature values (20 °C).

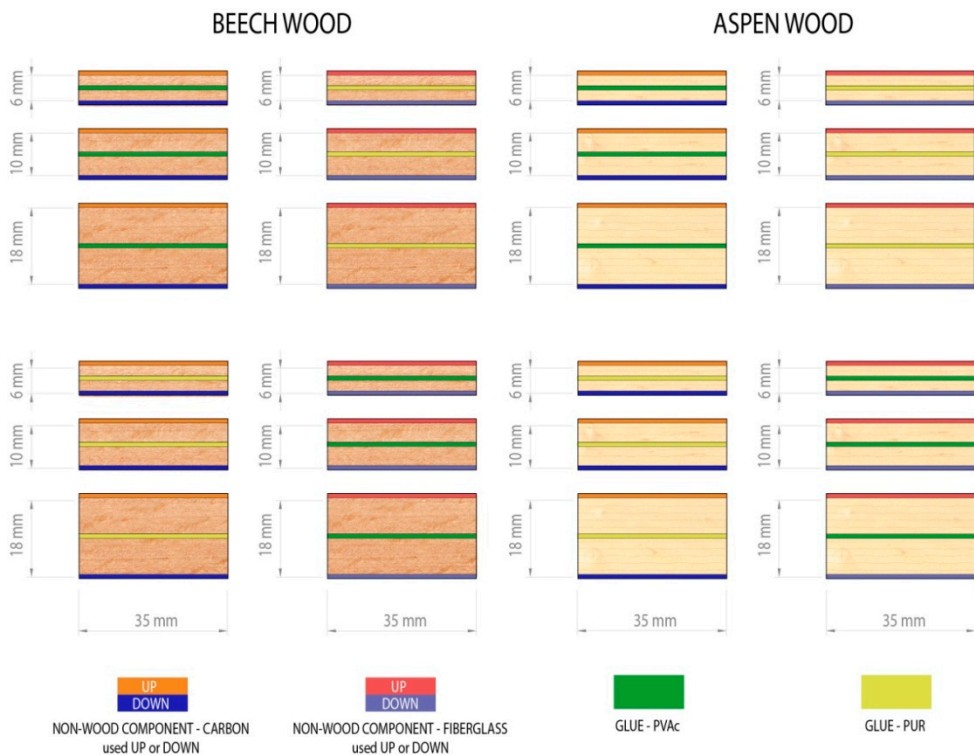

**Figure 2.** Categorization of the test specimens.

**Table 1.** Parameters of the polyvinyl acetate adhesive (PVAc) and polyurethane adhesive (PUR) Adhesives.

| Technical Data | AG-COLL 8761/L D3 | NEOPUR 2238R |
|---|---|---|
| Viscosity (mPa) | 5000 to 7000 by 23 °C | 2000 to 4500 by 25 °C |
| Working Time (min) | 15 to 20 | 60 |
| Density (g/cm$^3$) | 0.9 to 1.1 by 23 °C | approx. 1.13 |
| NCO Content (%) | - | approx. 15.5 to 16.5 |
| Color | White, milk | Brown |
| Open Time (min) | 15 | approx. 20 to 25 |
| Dry Matter Content | 49 to 51 | 100 |
| pH | 3.8 to 4.5 | - |

*2.2. Methods*

2.2.1. Determining Selected Characteristics

The three-point bending test was performed according to the predetermined conditions. The lower support span was set to 20 times the thickness of the test specimen (this span varied, depending on the thickness of the test specimens). The bending tests were performed on a universal testing machine (FPZ 100, TIRA, Schalkau, Germany), according to [35–37]. To adhere to the duration limit for the test (2 min.), the top support feed rate was set to 3 mm/min. The deflections at the limit of proportionality and at the modulus of rupture were measured using an ALMEMO 2690-8 datalogger (Ahlborn GmbH, Braunschweig, Germany).

A force-deflection diagram was used to determine the limit of proportionality and the modulus of rupture. To identify all of the necessary characteristics, a program was developed for identifying data that can be obtained from the force-deflection diagram.

2.2.2. Evaluation and Calculation

To determine the influence of the sample type on the bending characteristics, an analysis of variance (ANOVA) and a Fischer's F-test were performed using Statistica 12 software (Statsoft Inc., Tulsa, OK, USA).

For the determination of deflection at the limit of proportionality and at the modulus of rupture, it was necessary to calculate the limit of proportionality and the modulus of rupture.

We calculate the limit of proportionality "*LOP*" in three point bending according to EN 310 [35] and Equation (1):

$$LOP = \frac{3F_E l_0}{2bh^2} \tag{1}$$

where *LOP* is the limit of proportionality of material (MPa), $F_E$ is force at the limit of proportionality (N), $l_0$ is the distance between supporting span (mm), *b* is width of test samples (mm), and *h* is the height (thickness) of the sample (mm).

The bending strength "modulus of rupture (*MOR*)" in three point bending was calculated in accordance with ISO 13061-3 [38] and Equation (2):

$$MOR = \frac{3F_{max} l_0}{2bh^2} \tag{2}$$

where *MOR* is (bending strength) of material (MPa), $F_{max}$ is maximum (breaking) force (N), $l_0$ is the distance between supporting span (mm), *b* is width of test samples (mm), and *h* is the height (thickness) of the sample (mm).

**3. Results and Discussion**

Tables 2 and 3 show the average value and the coefficients of variation for the deflection at the limit of proportionality ($Y_E$) and deflection at the modulus of rupture ($Y_P$) in the laminated aspen and beech materials reinforced with a non-wood component (glass and carbon fiber) that is glued with PUR and PVAc adhesives. For the aspen wood (Table 2), the highest deflection at the limit of proportionality (4.89 mm) was measured on the 18-mm-thick test specimens that were glued with PVAc adhesive and reinforced with carbon fibers on the underside of the loaded specimen with respect to the loading direction. The greatest deflection at the modulus of rupture (16.70 mm) was measured on the 18-mm-thick laminated material that was glued with PUR adhesive and reinforced with glass fibers placed on the underside with respect to the loading direction. A comparison with the results of Sikora et al. verified the effect of the use of a non-wood material reinforcement to improve the deflection characteristics [39], and it was concluded that the use of a non-wood component significantly affected the deflection characteristics.

**Table 2.** Mean Values of the Deflection at the Limit of Proportionality and the Deflection at the Modulus of Rupture, and Coefficient of Variance of the Aspen Wood.

| WS | NWC | NWC Location | Glue | *T* (mm) | Code | $Y_E$ (mm) | $Y_P$ (mm) |
|----|-----|--------------|------|----------|------|------------|------------|
| A | CA | U | PUR | 6 | A-CA-U-PUR-6 | 1.59 (14.0) | 3.00 (18.1) |
| A | CA | U | PUR | 10 | A-CA-U-PUR-10 | 2.06 (16.8) | 8.01 (16.6) |
| A | CA | U | PUR | 18 | A-CA-U-PUR-18 | 2.45 (13.0) | 9.32 (13.2) |
| A | CA | U | PVAc | 6 | A-CA-U-PVAc-6 | 2.10 (9.0) | 7.97 (3.3) |
| A | CA | U | PVAc | 10 | A-CA-U-PVAc-10 | 2.81 (17.7) | 11.32 (5.5) |
| A | CA | U | PVAc | 18 | A-CA-U-PVAc-18 | 3.29 (21.0) | 12.24 (16.8) |
| A | LA | U | PUR | 6 | A-LA-U-PUR-6 | 1.98 (17.9) | 7.46 (14.4) |
| A | LA | U | PUR | 10 | A-LA-U-PUR-10 | 2.73 (15.4) | 7.74 (15.9) |
| A | LA | U | PUR | 18 | A-LA-U-PUR-18 | 3.81 (8.0) | 8.74 (17.1) |
| A | LA | U | PVAc | 6 | A-LA-U-PVAc-6 | 2.28 (5.7) | 9.07 (13.6) |
| A | LA | U | PVAc | 10 | A-LA-U-PVAc-10 | 2.70 (7.7) | 9.68 (19.5) |
| A | LA | U | PVAc | 18 | A-LA-U-PVAc-18 | 3.32 (16.5) | 9.90 (11.0) |
| A | CA | D | PUR | 6 | A-CA-D-PUR-6 | 1.54 (20.2) | 5.11 (19.4) |
| A | CA | D | PUR | 10 | A-CA-D-PUR-10 | 2.97 (7.4) | 9.07 (19.9) |
| A | CA | D | PUR | 18 | A-CA-D-PUR-18 | 3.22 (16.4) | 13.30 (16.9) |
| A | CA | D | PVAc | 6 | A-CA-D-PVAc-6 | 3.82 (15.0) | 8.46 (20.4) |
| A | CA | D | PVAc | 10 | A-CA-D-PVAc-10 | 4.09 (20.6) | 12.95 (6.9) |
| A | CA | D | PVAc | 18 | A-CA-D-PVAc-18 | 4.89 (19.1) | 14.48 (17.9) |
| A | LA | D | PUR | 6 | A-LA-D-PUR-6 | 3.21 (18.5) | 8.23 (19.3) |
| A | LA | D | PUR | 10 | A-LA-D-PUR-10 | 3.56 (7.7) | 10.33 (16.4) |
| A | LA | D | PUR | 18 | A-LA-D-PUR-18 | 4.93 (8.8) | 16.70 (18.7) |
| A | LA | D | PVAc | 6 | A-LA-D-PVAc-6 | 3.48 (9.5) | 9.21 (6.2) |
| A | LA | D | PVAc | 10 | A-LA-D-PVAc-10 | 4.05 (12.9) | 10.51 (16.3) |
| A | LA | D | PVAc | 18 | A-LA-D-PVAc-18 | 4.22 (14.3) | 13.62 (21.1) |

WS—wood species; NWC—non-wood component, *T*—thickness, $Y_E$—deflection at the limit of proportionality, $Y_P$—Deflection at the modulus of rupture, A—aspen, CA—carbon, LA—fiberglass, U—up, and D—down

**Table 3.** Mean Values of the Deflection at the Limit of Proportionality and the Deflection at the Modulus of Rupture, and Coefficient of Variance of the Beech Wood.

| WS | NWC | NWC Location | Glue | *T* (mm) | Code | $Y_E$ (mm) | $Y_P$ (mm) |
|----|-----|--------------|------|----------|------|------------|------------|
| B | CA | U | PUR | 6 | B-CA-U-PUR-6 | 2.11 (20.1) | 5.13 (17.6) |
| B | CA | U | PUR | 10 | B-CA-U-PUR-10 | 2.22 (14.4) | 6.81 (19.8) |
| B | CA | U | PUR | 18 | B-CA-U-PUR-18 | 2.45 (8.7) | 7.38 (8.8) |
| B | CA | U | PVAc | 6 | B-CA-U-PVAc-6 | 2.59 (19.8) | 7.85 (9.3) |
| B | CA | U | PVAc | 10 | B-CA-U-PVAc-10 | 2.87 (8.1) | 9.48 (9.7) |
| B | CA | U | PVAc | 18 | B-CA-U-PVAc-18 | 3.48 (14.4) | 10.39 (9.5) |
| B | LA | U | PUR | 6 | B-LA-U-PUR-6 | 1.85 (12.6) | 5.42 (10.3) |
| B | LA | U | PUR | 10 | B-LA-U-PUR-10 | 2.85 (18.5) | 7.80 (17.0) |
| B | LA | U | PUR | 18 | B-LA-U-PUR-18 | 4.39 (18.7) | 9.25 (16.8) |
| B | LA | U | PVAc | 6 | B-LA-U-PVAc-6 | 2.37 (20.2) | 7.14 (4.8) |
| B | LA | U | PVAc | 10 | B-LA-U-PVAc-10 | 3.20 (17.6) | 10.25 (11.2) |
| B | LA | U | PVAc | 18 | B-LA-U-PVAc-18 | 4.03 (7.6) | 11.69 (19.3) |
| B | CA | D | PUR | 6 | B-CA-D-PUR-6 | 2.83 (19.5) | 7.26 (8.4) |
| B | CA | D | PUR | 10 | B-CA-D-PUR-10 | 3.04 (19.6) | 7.72 (14.9) |
| B | CA | D | PUR | 18 | B-CA-D-PUR-18 | 7.07 (20.1) | 15.12 (15.7) |
| B | CA | D | PVAc | 6 | B-CA-D-PVAc-6 | 3.00 (19.0) | 8.99 (12.9) |
| B | CA | D | PVAc | 10 | B-CA-D-PVAc-10 | 3.71 (16.1) | 10.07 (18.9) |
| B | CA | D | PVAc | 18 | B-CA-D-PVAc-18 | 6.69 (19.0) | 12.70 (16.1) |
| B | LA | D | PUR | 6 | B-LA-D-PUR-6 | 4.36 (17.9) | 11.38 (7.4) |
| B | LA | D | PUR | 10 | B-LA-D-PUR-10 | 5.33 (20.5) | 12.45 (17.4) |
| B | LA | D | PUR | 18 | B-LA-D-PUR-18 | 5.63 (12.4) | 15.33 (15.6) |
| B | LA | D | PVAc | 6 | B-LA-D-PVAc-6 | 2.87 (20.7) | 7.52 (8.1) |
| B | LA | D | PVAc | 10 | B-LA-D-PVAc-10 | 4.07 (19.3) | 11.84 (17.9) |
| B | LA | D | PVAc | 18 | B-LA-D-PVAc-18 | 6.77 (19.7) | 20.01 (18.9) |

WS—wood species, NWC—non-wood component, *T*—thickness, $Y_E$—deflection at the proportionality limit, $Y_P$—deflection at the modulus of rupture, B—beech, CA—carbon, LA—fiberglass, U—up, and D—down.

Table 3 shows the deflection characteristics ($Y_E$ and $Y_P$) of the layered beech materials that were reinforced with non-wood components. The greatest deflection at the limit of proportionality (7.07 mm) was measured on the 18-mm-thick test specimens that were bonded with PUR adhesive and reinforced with carbon fibers placed on the underside of the loaded test specimen relative to the loading direction. In the case of deflection at the modulus of rupture, the highest values (20.01 mm) were measured on the 18-mm-thick test specimens that were glued with PVAc adhesive and reinforced with glass fibers

placed on the underside with respect to the loading direction. The effect of the adhesive was verified while using the results of Gáborík et al. [40], who presented results of the deflection of beech lamellas that were glued using PUR and PVAc adhesives. Regarding the effect of using a non-wood component to reinforce the layered material, the results of the present study were compared with the results of Sikora et al. [39], with the same conclusion as for the layered aspen materials.

The measured data were statistically evaluated using one-factor analyses (ANOVA), where the factor was the type of test specimen. The code of test samples includes the wood species that are used in the laminated material, type of non-wood component and its position in the structure relative to the loading direction, type of adhesive, and total thickness of the test sample. The evaluation was based on the *p* significance level, which was *p* = 0.005. Table 4 shows the results of the statistical evaluation of the effect of the aspen and beech test sample type on the deflection at the limit of proportionality and deflection at the modulus of rupture of the laminated materials with a non-wood component on the underside and topside. It was clear from these results that the type of test sample had a significant effect on the deflection at the limit of proportionality and on the deflection at the modulus of rupture.

From Figure 3 and Table 4, it was clear that it was more beneficial to apply the reinforcing material to the down side of the laminated aspen material for the deflection at the limit of proportionality. Higher values were achieved in all of the cases, except one, where the values were almost equal. The values of the reinforced material glued on the down side were approximately 41.32% higher than the average values when it was glued on the top. When evaluating the material that was used for reinforcing solid wood, the average values with glass fiber were 15.62% higher than those measured with carbon fiber. The last important factor was the adhesive used to connect the individual elements together. The average values measured using PVAc adhesive were 20.56% higher than those measured with PUR adhesive.

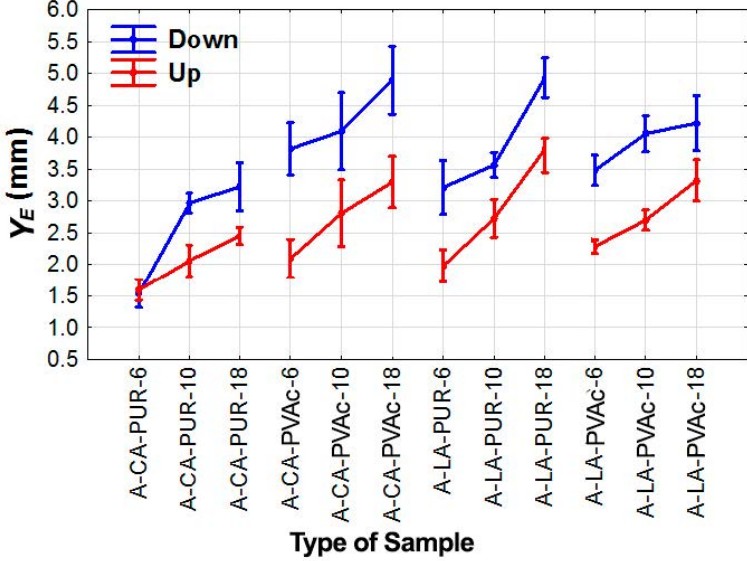

**Figure 3.** Effect of the aspen wood species on the deflection at the limit of proportionality.

The ANOVA test results in Figure 4 and Table 4 confirmed the same trend in the deflection at the limit of proportionality in the reinforced laminated beech material, as in the deflection at the limit of proportionality in the laminated aspen material. Higher values were achieved when the reinforcing material was glued on the down side. The average values in this case were up to 60.91% higher. The reinforcing materials were evaluated in the same way. The glass fiber had 13.46% higher deflection values than the carbon fiber. As for the adhesive used, there were no major differences. Expressed as a percentage difference of the mean values, the difference between PVAc and PUR was approximately 3.5%, where PVAc had higher deflection values at the limit of proportionality.

**Table 4.** Statistical Evaluation of the Effect of Type of Samples on the Deflection at the Limit of Proportionality and Deflection at the Modulus of rupture.

| Deflection at the Limit of Proportionality of Aspen and NWC Down | | | | | |
|---|---|---|---|---|---|
| **Monitored Factor** | **Sum of Squares** | **Degree of Freedom** | **Variance** | **Fisher's F-test** | **Significance Level *p*** |
| Intercept | 1469.556 | 1 | 1469.556 | 5410.635 | *** |
| (1) Type of Sample | 76.586 | 11 | 6.962 | 25.634 | *** |
| Error | 29.333 | 108 | 0.272 | | - |
| Deflection at the Limit of Proportionality of Aspen and NWC UP | | | | | |
| **Monitored Factor** | **Sum of Squares** | **Degree of Freedom** | **Variance** | **Fisher's F-test** | **Significance Level *p*** |
| Intercept | 469.9151 | 1 | 469.9151 | 5033.475 | *** |
| (1) Type of Sample | 19.7016 | 11 | 1.7911 | 19.185 | *** |
| Error | 9.4292 | 101 | 0.0934 | | - |
| Deflection at the Modulus of Rupture of Aspen and NWC Down | | | | | |
| **Monitored Factor** | **Sum of Squares** | **Degree of Freedom** | **Variance** | **Fisher's F-test** | **Significance Level *p*** |
| Intercept | 2373.329 | 1 | 2373.329 | 2497.682 | *** |
| (1) Type of Sample | 298.265 | 11 | 27.115 | 28.536 | *** |
| Error | 102.623 | 108 | 0.950 | | - |
| Deflection at the Modulus of Rupture of Aspen and NWC Up | | | | | |
| **Monitored Factor** | **Sum of Squares** | **Degree of Freedom** | **Variance** | **Fisher's F-test** | **Significance Level *p*** |
| Intercept | 5186.339 | 1 | 5186.339 | 4235.214 | *** |
| (1) Type of Sample | 288.155 | 11 | 26.196 | 21.392 | *** |
| Error | 139.602 | 114 | 1.225 | | - |
| Deflection at the Limit of Proportionality of Beech and NWC Down | | | | | |
| **Monitored Factor** | **Sum of Squares** | **Degree of Freedom** | **Variance** | **Fisher's F-test** | **Significance Level *p*** |
| Intercept | 13,863.51 | 1 | 13,863.51 | 3716.494 | *** |
| (1) Type of Sample | 1281.94 | 11 | 116.54 | 31.242 | *** |
| Error | 402.87 | 108 | 3.73 | | - |
| Deflection at the Limit of Proportionality of Beech and NWC Up | | | | | |
| **Monitored Factor** | **Sum of Squares** | **Degree of Freedom** | **Variance** | **Fisher's F-test** | **Significance Level *P*** |
| Intercept | 6405.696 | 1 | 6405.696 | 4427.627 | *** |
| (1) Type of Sample | 373.911 | 11 | 33.992 | 23.495 | *** |
| Error | 146.122 | 101 | 1.447 | | - |
| Deflection at the Modulus of Rupture of Beech and NWC Down | | | | | |
| **Monitored Factor** | **Sum of Squares** | **Degree of Freedom** | **Variance** | **Fisher's F-test** | **Significance Level *P*** |
| Intercept | 15,277.64 | 1 | 15,277.64 | 4198.613 | *** |
| (1) Type of Sample | 1739.17 | 11 | 158.11 | 43.451 | *** |
| Error | 392.98 | 108 | 3.64 | | - |
| Deflection at the Modulus of Rupture of Beech and NWC Up | | | | | |
| **Monitored Factor** | **Sum of Squares** | **Degree of Freedom** | **Variance** | **Fisher's F-test** | **Significance Level *P*** |
| Intercept | 5186.339 | 1 | 5186.339 | 4235.214 | *** |
| (1) Type of Sample | 288.155 | 11 | 26.196 | 21.392 | *** |
| Error | 139.602 | 114 | 1.225 | | - |

NS—not significant, ***—significant at $p < 0.005$.

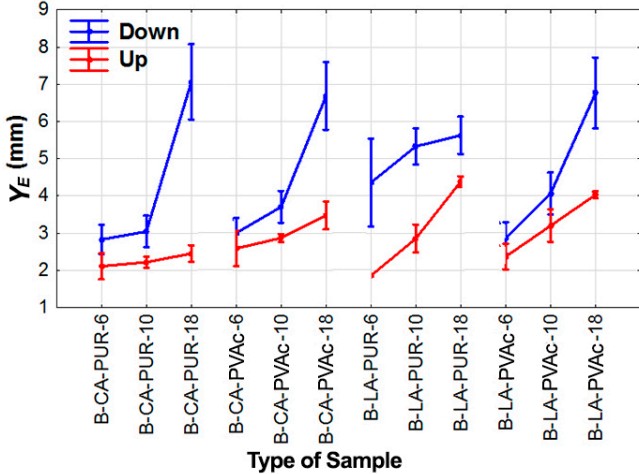

**Figure 4.** Effect of the beech wood species on the deflection at the limit of proportionality.

Figure 5 graphically depicts a comparison of the factors that were focused on the deflection at the modulus of rupture in the aspen wood. The dependencies were similar to those of the deflection at the limit of proportionality, as seen in Table 4. A greater deflection was observed when the reinforcing material was glued opposite to the loaded side (by 26.35% in this case). The average deflection values were 5.17% higher when glass fiber. The adhesive used was the last important factor. With PVAc glue, the deflection at the modulus of rupture was 20.93% higher.

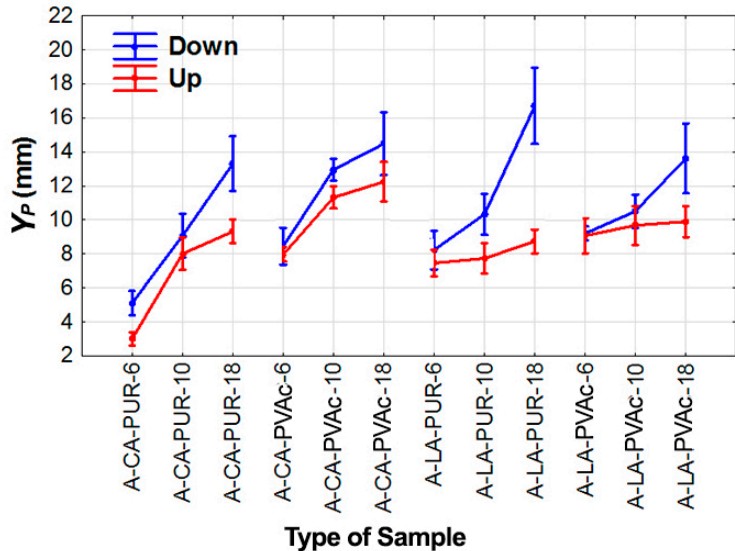

**Figure 5.** Effect of the aspen wood species on the deflection at the modulus of rupture.

As for all of the previous cases, higher deflection at the modulus of rupture values was achieved in the reinforced laminated beech materials by gluing the reinforcing material to the down side relative to the loading direction (Figure 6). These values were approximately 36% higher than these measured in the materials with the non-wood component glued on the top (Table 4). With glass fibers, the deflection at the modulus of rupture values increased by approximately 20% when compared with carbon fibers. A further increase in the deflection values can be achieved by using the correct adhesive; in this study, the deflection values for the samples with PVAc were 15.2% higher.

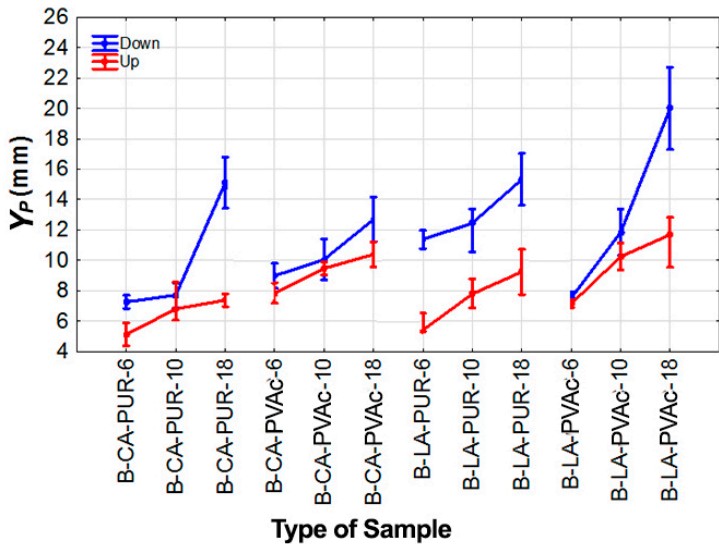

**Figure 6.** Effect of the beech wood species on the deflection at the modulus of rupture.

## 4. Conclusions

1.  For the laminated aspen and beech materials that were reinforced with non-wood components, the lowest deflection values were measured in the test specimens with the non-wood component on the up side with respect to the loading direction, and the highest deflection values were measured in the test specimens with the non-wood component on the down side with respect to the loading direction.

2.  Glass fiber was proven to be better non-wood component for both the laminated beech and aspen materials with regard to the deflection at the limit of proportionality and at the modulus of rupture in terms of the non-wood component itself.

3.  In the case of the adhesive used, it cannot be clearly stated which adhesive was the most effective. However, higher average values of deflection at the limit of proportionality and at the modulus of rupture were measured with the PVAc adhesive.

4.  As was expected, the thickness of the material proved to be an important factor that affected the deflection at both the limit of proportionality and at the modulus of rupture.

5.  In the case of aspen composition for better bendability, we suggest gluing with PUR adhesive and reinforcement with glass fiber on the down side with respect to the loading direction. In addition, in the case of beech composition, we suggest gluing with PVAc adhesive and reinforcement with glass fiber on the down side with respect to the loading direction.

**Author Contributions:** Resources, A.S., T.S., V.Z. and Z.G.; Supervision, A.S.; Visualization, T.S.; Writing—original draft, A.S. and T.S.; Writing—review & editing, A.S. and T.S.

**Funding:** The authors are grateful for the support of the Advanced Research Supporting the Forestry and Wood-processing Sector's Adaptation to Global Change and the 4[th] Industrial Revolution (Project No. CZ.02.1.01/0.0/0.0/16_019/0000803), financed by OP RDE. The authors are also grateful for the support of the Internal Grant Agency (IGA) of the Faculty of Forestry and Wood Sciences (Project No. B 06/17).

**Conflicts of Interest:** The authors declare no conflict of interest.

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
