# Peer review of "Effect of Selected Factors on the Bending Deflection at the Limit of Proportionality and at the Modulus of Rupture in Laminated Veneer Lumber"

_forests, doi:10.3390/f10050401_

Round 1

Reviewer 1 Report

The paper is written well, following a logical consistent style. Despise I am not competent to evaluate its language quality; this seems fully appropriate.

My comments to the content are the following:

1.      The text body displays some coherence gaps. It seems glued-together of parts of several just published papers.

2.      The tables are too numerous (perhaps redundant), with vast numbers of not easily readable data.

3.      The results of factorial analysis should be summarised in one table (if any)

4.      The tables with the results of the Duncan test are, according to my opinion, redundant. They do not seem attractive for a detailed reading. Moreover, the information required is provided in Figs 3 to 6.

5.      I have also objections against the references. The works referred do not seem much connected with the issue discussed in the paper. On the other hand, there are lacking works dealing with wood reinforced with carbon fibres.

6.      The paper contents several imprecisions such as: state is to replace by shape – line 38, misplaced parentheses with data – line 116. The most serious slip-up seems inconsistent use of upper/lower and convex/concave in the text and the terms of yield point in the text and modulus of rupture in Fig. 1.

Author Response

Response to Reviewer 1 Comments

Point 1: The text body displays some coherence gaps. It seems glued-together of parts of several just published papers.

 Response 1: Please specify the gaps, so we can fix them.

Point 2: The tables are too numerous (perhaps redundant), with vast numbers of not easily readable data.

 Response 2: Table was there for more detailed analysis, but we agree with you that they are not needed. All necessary result are shown at figure 3 to 6.

Point 3: The results of factorial analysis should be summarised in one table (if any)

 Response 3: Here we think that for the sake of clarity in the analysis is best solution of several smaller tables.

Point 4: The tables with the results of the Duncan test are, according to my opinion, redundant. They do not seem attractive for a detailed reading. Moreover, the information required is provided in Figs 3 to 6.

 Response 4: Table was there for more detailed analysis, but we agree with you that they are not needed. All necessary result are shown at figure 3 to 6.

Point 5: I have also objections against the references. The works referred do not seem much connected with the issue discussed in the paper. On the other hand, there are lacking works dealing with wood reinforced with carbon fibres.

 Response 5: References were added.

Point 6: The paper contents several imprecisions such as: state is to replace by shape – line 38, misplaced parentheses with data – line 116. The most serious slip-up seems inconsistent use of upper/lower and convex/concave in the text and the terms of yield point in the text and modulus of rupture in Fig. 1.

 Response 6:   Line 38 was fixed.

                        Line 116 was fixed.

                        Terms upper/lower and convex/concave was unified.              

                        Term yield point was changed on modulus of rupture.

Reviewer 2 Report

Review report for the manuscript: Forests-476655

The manuscript analyses the bending characteristics of LVL reinforced by glass and carbon fibers. The variation of the thickness of veneer used glue and the position of the reinforcement is additionally included in the study. The topic is generally important and relevant; however, the manuscript needs some qualitative changes.

Introduction

The information related to the reinforcing of wood, and especially LVL, with synthetic fibers and achieved results is missing. No, or very low discussion on results related to other studies is present.

L44

"Deflection at the yield point is characterized as the moment when the material breaks" This is not the yield point, but the deformation at maximum load. Please explain the method you defined the yield point or the reference!

L74

The static mechanical testing according to the information followed the EN310 standard. The standard suggests conditioning of samples in normal climate! Why the 8% humidity was used?

L78

The figure is blurry. There is also not enough accurately presented, what was the composition of LVL. Please explain the used of different colors in-between veneer layers!

L87

There is no definition of limit of proportionality, as well as yield point in the standards (EN 310), which were used in the research (ref 20, 21, 22). Please define clearlly, what was the principle to determine these two variables!

Table 4 to 18

There is a need to explain some confused interpretation of the results and used analysis. It is obvious, that ANOVA was made according to the 12 different samples, connected to the composition of the LVL (figure 2). However, from the figure 2, it looks there were just 4 types of composition, where additionally the thickness of the veneer varied. This is clear also from the presentation of the result on figures 3 to 6.

Is there really a need to present Tables 12 to 19, where you are explaining what are the differences related to the used veneer thickness, combined with other variables (synthetic fibers, glue, wood species). It is suggested to add the correlation coefficient, related to the veneer thickness already in the figures (3 to 6), which will lead to reduce the sizes of tables 12 to 19. There will be no loose of the information.

However, the figures 3 to 6, have no additional information, related to table 2. It looks like there are redundant!

L244 – Conclusions

There is no clear explanation what is the positive or negative effect of adding fibrous reinforcement into LVL. If the goal was to improve the bendability of LVL, then explain please what is the optimum reinforcement or the composition of the LVL, you suggest?

Author Response

Response to Reviewer 2 Comments

Point 1: The manuscript analyses the bending characteristics of LVL reinforced by glass and carbon fibers. The variation of the thickness of veneer used glue and the position of the reinforcement is additionally included in the study. The topic is generally important and relevant; however, the manuscript needs some qualitative changes.

 Introduction

 The information related to the reinforcing of wood, and especially LVL, with synthetic fibers and achieved results is missing. No, or very low discussion on results related to other studies is present.

 Response 1: Information was added.

Point 2: L44 - "Deflection at the yield point is characterized as the moment when the material breaks" This is not the yield point, but the deformation at maximum load. Please explain the method you defined the yield point or the reference!

 Response 2: Deflection on the yield point was changed on the defection at the modulus of rupture.

 Point 3: L74 - The static mechanical testing according to the information followed the EN310 standard. The standard suggests conditioning of samples in normal climate! Why the 8% humidity was used?

 Response 3: Thank you for the warning, it was a mistake in the text. The test specimens were air conditioned to 12% humidity at relative humidity of 65% and temperature of 20 °C.

Point 4: L78 - The figure is blurry. There is also not enough accurately presented, what was the composition of LVL. Please explain the used of different colors in-between veneer layers!

 Response 4: We think that quality of figure is in adequate quality. Between layers are marked different type of adhesive as you can see in legend of this figure.

Point 5: L87 - There is no definition of limit of proportionality, as well as yield point in the standards (EN 310), which were used in the research (ref 20, 21, 22). Please define clearlly, what was the principle to determine these two variables!

 Response 5: Definition was added in methodology.

Point 6: Table 4 to 18 - There is a need to explain some confused interpretation of the results and used analysis. It is obvious, that ANOVA was made according to the 12 different samples, connected to the composition of the LVL (figure 2). However, from the figure 2, it looks there were just 4 types of composition, where additionally the thickness of the veneer varied. This is clear also from the presentation of the result on figures 3 to 6.

 Response 6: Overall number of test group was 48 (information was added into material section) with respect to the thickness and species of wood component, the type of non-wood component and the position of the non-wood component relative to the load direction. From figure 3-6 is clear (information in horizontal axis and different color) that we had 48 different test group. In case of figure 2, picture would be very large if we should draw every single test group individually, so we decided to combine them and in legend of picture you can see all necessary information for understanding.

Point 7: Is there really a need to present Tables 12 to 19, where you are explaining what are the differences related to the used veneer thickness, combined with other variables (synthetic fibers, glue, wood species). It is suggested to add the correlation coefficient, related to the veneer thickness already in the figures (3 to 6), which will lead to reduce the sizes of tables 12 to 19. There will be no loose of the information.

However, the figures 3 to 6, have no additional information, related to table 2. It looks like there are redundant!

 Response 7: Table are there for more detailed analysis, but we agree with you that they are not needed. All necessary result are shown at figure 3 to 6 and table 2 and 3. Figure 3 to 6 is there for better visualization of results.

Point 8: L244 – Conclusions

There is no clear explanation what is the positive or negative effect of adding fibrous reinforcement into LVL. If the goal was to improve the bendability of LVL, then explain please what is the optimum reinforcement or the composition of the LVL, you suggest?

 Response 8: Positive effect of reinforcement with synthetic fiber is included in introduction. The conclusion was extended with the required information.

Round 2

Reviewer 2 Report

Review report for the manuscript: Forests-476655-rev2

The manuscript analyses the bending characteristics of LVL reinforced by glass and carbon fibers. The variation of the thickness of veneer used glue and the position of the reinforcement is additionally included in the study. The topic is generally important and relevant; however, the manuscript still contains some shortcomings that need to be corrected.

Abstract

I propose that at the end of the abstract you should clearly outline the main findings of the study. The current description “The results of the study are of great importance…”is too general!

Material and Methods

L83 "The test specimens were climatized at a 12% humidity…” The term humidity is suggested to be changed into moisture content (MC) or even equilibrium moisture content (EMC), which was actually reached by conditioning at 20°C/65 RH.

Results and Discussion

L143 In the title (also in the legend bellow) is Yp denoted as Modulus of rupture (MOR). However, according to the figure 1, it is a deflection at the MOR. Please correct also in the following text.

L160 There is still old denotation of the Yp (deflection at the yield point)!?! Please correct is, as well as in the whole manuscript!

L162 “The measured data was statistically evaluated using one-factor analyses” Please check the grammar, else elsewhere in the text. Please explain what is one-factor analysis? Isn’t just ANOVA?

L168 Explaining and the discussion of the results from statistical evaluation (Table 4 to 11) is too general. There are several statements, i.e. “It was clear from these result that the test specimen type had a significant effect…” It is suggested to define these effects clearly – where Ye and Yp increased or decreased? Is this good or bad? This might be improved also with the following discussion (L221-), where some referencing to the tables 4 to 11 is suggested.

L264 – Conclusions

“In terms of the non-wood component itself, glass fiber was proven to be a better non-wood component for both the laminated beech and aspen materials with regard to the deflection at the limit of proportionality and the modulus of rupture.” Please check the grammar! The statement says at the moment, that MOR was also determined, and not just the deflection at the MOR! The same formulation has been found elsewhere in the text!

Author Response

2nd response to Reviewer 2 Comments 

 Point 1: I propose that at the end of the abstract you should clearly outline the main findings of the study. The current description “The results of the study are of great importance…”is too general!

Response 1: Abstract was improved.

Point 2: L83 "The test specimens were climatized at a 12% humidity…” The term humidity is suggested to be changed into moisture content (MC) or even equilibrium moisture content (EMC), which was actually reached by conditioning at 20°C/65 RH.

Response 2: OK, The term was changed in Article. Thank you for correction.

Point 3: L143 In the title (also in the legend bellow) is Yp denoted as Modulus of rupture (MOR). However, according to the figure 1, it is a deflection at the MOR. Please correct also in the following text.

Response 3: In the title of table is written Deflection at he LOP and MOR, we have changed it to " Deflection at the Limit of Proportionality and the Deflection at the Modulus of Rupture" for a clearer explanation. In the legend is our mistake - It is corrected now.

Point 4: L160 There is still old denotation of the Yp (deflection at the yield point)!?! Please correct is, as well as in the whole manuscript!

Response 4: OK, I am sorry, corrected.

Point 5: L162 “The measured data was statistically evaluated using one-factor analyses” Please check the grammar, else elsewhere in the text. Please explain what is one-factor analysis? Isn’t just ANOVA?

Response 5: Grammary was checked, Yes, one-factor analyses is single factorial ANOVA.

Point 6: L168 Explaining and the discussion of the results from statistical evaluation (Table 4 to 11) is too general. There are several statements, i.e. “It was clear from these result that the test specimen type had a significant effect…” It is suggested to define these effects clearly – where Ye and Yp increased or decreased? Is this good or bad? This might be improved also with the following discussion (L221-), where some referencing to the tables 4 to 11 is suggested.

Response 6: References to the tables 4 to 11 was added.

Point 7: L264 – Conclusions

“In terms of the non-wood component itself, glass fiber was proven to be a better non-wood component for both the laminated beech and aspen materials with regard to the deflection at the limit of proportionality and the modulus of rupture.” Please check the grammar! The statement says at the moment, that MOR was also determined, and not just the deflection at the MOR! The same formulation has been found elsewhere in the text!

Response 7: Statement was corrected: In terms of the non-wood component itself, glass fiber was proven to be better non-wood component for both the laminated beech and aspen materials with regard to the deflection at the limit of proportionality and at the modulus of rupture.